# Bending and Crack Evolution Behaviors of Cemented Soil Reinforced with Surface Modified PVA Fiber

**DOI:** 10.3390/ma15144799

**Published:** 2022-07-08

**Authors:** Lisheng Liang, Yaxing Xu, Shunlei Hu

**Affiliations:** 1Department of Civil and Architectural Engineering, Shanxi Institute of Technology, Yangquan 045000, China; 2The Cultivation Base of Shanxi Key Laboratory of Mining Area Ecological Restoration and Solid Wastes Utilization, Yangquan 045000, China; 3College of Civil Engineering, Taiyuan University of Technology, Taiyuan 030024, China; star_xu@tom.com (Y.X.); hushunlei97@163.com (S.H.)

**Keywords:** PVA fiber, surface modifications, flexural properties, crack propagation behavior, DIC

## Abstract

To improve the flexural properties of cemented soils reinforced with fibers and avoid their brittle failure when subjected to complex loading conditions, a simple and cost-effective technique was explored to facilitate their application in retaining walls. In this study, how different fiber surface modifications, i.e., alkali treatment, acid treatment and silane coupling agent treatment, as well as different fiber contents, i.e., 0%, 0.25%, 0.5% and 1%, affect the bending properties of cemented soils was investigated by conducting three-point bending tests on notched beams. The digital image correlation (DIC) technology was used to examine the crack propagation process and the strain field distribution of cracks in specimens in the flexural tests. The results show that all fiber surface modifications increased peak strength and fracture energy, for example, the fracture energy of specimens AN1, AH1 and AK1 was increased by 180.4%, 121.5% and 155.4%, respectively, compared to PVA1. In addition, the crack tip strain, crack propagation rate and the initial crack width of the modified specimens were lower than those before modification. Lastly, scanning electron microscope (SEM) and mercury intrusion porosimetry tests were adopted to reveal the mechanism of bending performance in cemented soils reinforced by fiber surface modifications.

## 1. Introduction

Cemented soils retaining wall are a typical type of foundation pit support in soft soil areas [1]. This conventionally includes 10–20% cement, and thus has some common shortcomings stemming from the use of cement, i.e., poor flexural properties, rapid crack propagation after cracking and a high risk of brittle failure [2,3]. Many researchers attempted to improve its mechanical properties by adding short fibers into cemented soils [4,5,6,7,8,9]. Polyvinyl alcohol (PVA) fibers show good potential in reinforce cemented soil due to their high strength, chemical corrosion resistance, good elasticity, wear resistance, and uniform dispersion [10,11]. However, the high cost of PVA fibers hinders their practical application, especially in China. According to the fiber bridging theory, fiber plays an important role in stress transfers of matrix, transferring the stress at the crack to the surrounding matrix [12]. The interfacial bonding between fibers and matrix is the key factor in the control of the high ductility of cement-based composites [13]. Exploring a simple, low-cost and more effective method to enhance the interfacial bonding between fibers and soil matrix is necessary [14]. 

Composites with chemical treatment show excellent mechanical properties [15,16,17,18]. Lopattananon et al. [19] studied the effects of different NaOH solution concentrations (1%, 3%, 5% and 7%) on the mechanical properties of pineapple leaf fiber-reinforced natural rubber composite. The results show that the tensile properties of fiber treated with 5% concentration increased by 28% compared with the unreinforced specimens. Yan et al. [20] investigated the effect of alkali (5%) treatment on the mechanical properties of coconut shell fiber-reinforced epoxy resin composite and found that the flexural strength and tensile strength of the composite increased by 16.7% and 17.8%, respectively. Nadia et al. [21] demonstrated that nitric acid increased the surface roughness of fibers, and consequently increased the surface tension by 40%, and significantly improved wettability. Shi et al. [22] performed the pull-out tests and concluded that the peak pull-out load of the fiber grafted with silane groups on the surface was three times that of the unmodified fiber, while the energy absorption increased by 121%.

Related research shows that fiber surface modification can increase the roughness of the fiber surface, improve the fiber surface energy, and achieve surface activation, thus increasing the force between the fiber and the matrix and improving the interfacial bonding properties between fiber and the matrix. However, most previous studies mostly focused on fiber surface-modification-reinforced epoxy resin [23,24] and cement matrix composites [25,26,27]. In addition, the research focus was mainly on the mechanical properties of fiber-modified-reinforced cementitious composites, and there are few reports on the crack propagation behavior during fracture and engineering applications; at the same time, there are few studies on the application of fiber modification to the improvement in the flexural properties of cement-retaining walls. 

Therefore, the aim of this study is to analyze the effects of different PVA fiber surface modification methods (i.e., alkali treatment, acid treatment, silane coupling agent treatment) and fiber content (0%, 0.25%, 0.5% and 1%) on the bending behavior and strain field properties during fracture of the cemented soil. Thirteen groups of specimens with different ratios were prepared, and parameters such as load-crack opening width curve, flexural strength and fracture energy of specimens were analyzed by three-point bending tests. In addition, the digital image correlation (DIC) technique was used to reproduce the crack propagation process of fiber-reinforced cemented soil, and a fracture energy prediction model was established using regression analysis, thus providing a theoretical reference for the engineering application of fiber-reinforced cemented soil.

## 2. Materials and Methods

### 2.1. Materials

The test soil was obtained from a construction site in Taiyuan, China. The index properties of the soil are given in Table 1.

PVA Kurary II type fiber produced by Kuraray Company in Japan, was used in tests. In the following sections, Kurary II type fiber is generally represented by PVA fiber. Each single fiber had a length of 12 mm. The index properties of the fiber are listed in Table 2. The cement used in the test was commercially available 42.5 grade ordinary silicate cement.

### 2.2. Fiber Surface Modification

Pretreating the PVA fiber is necessary to remove the residues on fiber surface, and thus to yield a better bonding between fiber and the surrounding matrix. Chun et al. [28] concluded that acetone washing could effectively improve the interfacial bond strength and fiber pullout energy. The same technique was also used in this study. Fibers were soaked in acetone solution for two hours. Then, they were washed three times with distilled water and dried at room temperature. The fiber surface was then optimized by applying the modification method detailed in Table 3. Once the modification was completed, fibers were washed again to a neutral condition. Eventually fibers were dried out in an oven at a temperature of 80° at least for 1 h.

### 2.3. Specimen Preparation

The retrieved soil specimens were dried, crushed and passed through a sieve with openings of 2 mm. The soil and cement were mixed together with the required dry masses to achieve targeting mass ratios, as needed in tests (Table 4). The modified fibers were then added into the cemented soils mixture; meanwhile, some water was also added to give slurries a mass water content of 33%. The slurries were fully stirred until a uniform state was attained. After that, the slurry was carefully transferred into a cubic mold with dimensions of 40 mm × 40 mm × 160 mm. The filled mold was then vibrated for about 40 s on a vibration table, during which the bubbles were possibly expelled. The surface of the vibrated specimens was then levelled using a scraper. After curing for 24 h, the specimens were removed from the mold. The specimens were then placed in a standard curing box with a relative humidity above 95% and a temperature of 20 ± 2 °C for 28 days. In order to investigate the effect of the fiber modification method and the fiber content on the bending performance of cemented soil, 13 groups of tests were performed, as shown in Table 4.

### 2.4. Three-Point Bending Test Method

After the cemented soil specimens were maintained at the required curing age, sandpaper was used to lightly polish the sides of the specimens. Lines were drawn at the bottom and both sides of the specimens, and a notch with a depth of 6 mm and a width of 2 mm was then made with a hand saw along the lines. The notch was straight and relatively complete when cutting.

According to the EN14651-2005 standard [29], the three-point bending test was carried out on a microcomputer-controlled electronic universal testing machine (range: 5 kN, accuracy: 0.1 N). The loading rate of the test was 0.2 mm/min, and the specimen dimensions are schematically shown in Figure 1.

### 2.5. Digital Image Correlation (DIC) Test Device

Bending strength and toughness are the main performance indexes used to study the bending behavior of cement-based materials; however, these do not reflect the bending failure process of the material. Digital image correlation (DIC) is an optical technique that can obtain the full-field displacement of the specimens surface and is widely used for crack monitoring due to its simplicity and high resolution [30,31,32]. In this paper, the DIC technique was used to study the crack propagation process in prefabricated notched beams. Firstly, white paint was sprayed on the middle part of the flat side and, after drying, the scattered speckle pattern was sprayed with black paint after the white paint was dried out, it was ensured that the speckle spots were of different sizes, hierarchical and clearly visible. The DIC image data acquisition equipment adopted a charge-coupled device (CCD) industrial camera with a resolution of 3840 × 2748, and the specimens were acquired at a frame rate of 7 frames/second. To obtain a uniform and easily processed image, a pair of LED light sources were placed on both sides of the test device. The test process was shot at an interval of 3 s until the specimen displacement of 4 mm was attained. Schematic diagram of DIC test system shown in Figure 2. The image post-processing software, GOM Correlate 2021 was used. During data processing, a measurement line was firstly drawn at the bottom of the specimens’ surface, as shown in Figure 3, to measure the opening width of the prefabricated cracks.

## 3. Results and Discussions

### 3.1. Bending Behavior

#### 3.1.1. Load-Crack Opening Width Curve

From Figure 4a–c, it can be seen that the load-crack opening width (P-CMOD) curves in the specimens mainly occur in three stages. The first stage is the elastic stage from the beginning of loading to the onset of cracks, during which fiber modification does not significantly affect the specimen behavior. This stage mainly depends on the properties of the cement matrix. The second stage is the non-linear propagation stage of the cracks. Compared with PC, the addition of certain number of fibers is able to effectively suppress the formation of original defects (i.e., interfaces and voids) as well as the propagation of microcracks in the specimens. When the modified fiber contents are 0.5% and 1%, the bridging effect of fibers is activated. The load increases to its peak (i.e., post-cracking load), and the P-CMOD curve shows an obvious strain-hardening response. The modified fibers improve due to the good interfacial effect of the modified fibers, which make it possible to transfer the load better, as a good interfacial effect exists and crack propagation is inhibited. When the specimens evolve to the residual stage, the load sharply decreases with the growth in the crack opening width. The load in modified fiber specimens decreases more slowly, perhaps because the fiber surface modification improves the microstructure of the matrix and increases the interfacial compactness. This may limit the increase in the crack width and lead to a slower decreasing curve. 

#### 3.1.2. Flexural Strength

It can be seen from Figure 5 that the initial cracking strength and peak strength of the modified fiber are approximately improved in comparison with unmodified fibers. When the fiber content is 0.25%, the modified fibers hardly affect the strength of the cemented soils. The initial crack strength is equal to the peak strength, because the fiber content is lower and the bridging effect of fibers is not fully triggered. When the fiber content approaches 0.5% and 1%, the bending strength of the modified fiber shows a second peak (which is higher than the first peak). When the fiber content is 1%, the peak strength of PN1, PH1 and PK1 increases by 73.3%, 61.7% and 49.9%, respectively, when compared with PVA1, which may result from the increased possibility of fiber distributing in the horizontal direction of the cracks. Moreover, the interface bonding effect between the modified fiber and the matrix is improved, which leads to better stress redistribution and transfer, and thus the fracture resistance is increased. The strength reduction is less pronounced and exceeds the initial crack strength. It can be seen that the rate of load drop and the strength recovery of the specimens are highly dependent on the orientations of fibers, as well as the amount of fibers in the cemented soil, which are bonded with the matrix.

#### 3.1.3. Fracture Energy

The fracture energy *G*_f_ is usually calculated as the ratio of the fracture work over the cross-sectional area of the fracture zone [33]. Figure 6 indicates that the fracture energy of the modified specimens was significantly increased by the presence of fibers. *G*_f_ is further increased once more fibers are included. When fiber content is 0.5%, the fracture energy of the fiber-modified specimens by NaOH, HCl and KH550 increases by 182.1%, 157.1% and 125%, respectively, in comparison with those specimens without modification. Specimens modified by NaOH were observed to have higher fracture energy. The increase in fracture energy is mainly caused by the stress transferred from the matrix to the fiber after cracking. The modified fiber has a micro-rough surface and greater hydrophilicity, yielding a better mechanical interlocking ability with the cement matrix; therefore, the modified fiber can better bridge the cracks and enhance the energy absorption capacity of the cemented soil.

### 3.2. Crack Propagation Behavior

#### 3.2.1. Strain Field on the Specimen Surface

Previous studies have shown that the bending behavior of specimens is closely related to the crack propagation process, especially after cracking, when fibers inhibit crack propagation by bridging the fracture surface [34,35]. Investigating the evolution of the strain field at the crack tip provides a potential way of better understanding this behavior.

By obtaining the strain field on the specimens’ surface through post-processing software, the crack evolution process in the specimens under bending stress includes approximately five stages, as illustrated in Figure 7.

(1)At the initial stage of loading, damage to the specimens’ surface was rarely observed.(2)The second stage was the microcrack initiation stage, during which no visible cracks on the specimens’ surface were observed, but some sudden strain changes at the crack tip became visible on the post-processing software.(3)The third stage was microcrack evolution stage when the applied load approaches the peak strength of the specimens and cracks were visible on the specimens’ surface.(4)The following stage was the macroscopic crack propagation stage, during which a significant increase in the length and width of the crack was shown.(5)In the final stage, the loading stopped and the crack had a considerable width and extended through the whole height of the specimen.

Figure 7 shows that the fracture failure characteristics of the specimens are more significant. The location and direction of the prefabricated notch determine the initiation point and the direction in which the crack propagates. The crack almost runs through the whole specimens when fracture failure occurs. The strain values at the crack tip of the modified specimens decreases when compared with those specimens without modification. For example, magnitude of the strain at the crack tip in the unmodified specimens was 1.026% at the microcrack initiation stage, while the strains were 0.751%, 0.659% and 0.845% respectively, for the specimens modified by NaOH, HCl and KH550. Modification by NaOH, HCl and KH550 caused a reduction of 26.8%, 35.7% and 17.6%, respectively, when compared with those specimens before modification. Modified fibers effectively inhibit the crack propagation in the specimens, thus improving the deformation resistance of the specimens.

#### 3.2.2. Strain Field at Crack Tip

As can be seen in Figure 7, the strain field around the crack of the specimens is zero at the beginning (green hue). As the loading time elapses, a red spike can be observed at the crack tip, which indicates that the strain is concentrated in this region. However, there is no strain at the center of the crack. The magnified image of the crack tip is shown in Figure 8. The area around the crack tip can be divided into four regions based on the strain field: (I) the compression region below the loading point (green region), (II) the fracture process zone (yellow zone), (III) the microcrack initiation stage region, and (IV) the macroscopic crack propagation region visible to the naked eye [36].

#### 3.2.3. Crack Morphology

As can be seen in Figure 9, the damage type of the unmodified cemented soil specimens is typical brittle fracture. The crack shows an almost perfect straight-line shape. The cracks in the modified specimens are deflected during the expansion process, and the path becomes increasingly curved. The notch tip is usually accompanied by a certain inclination of fine cracks, indicating that the crack tip encounters a great resistance during the expansion process, which will most likely benefit from the strong interlocking effect and the interface friction between modified fiber and cemented soil. The development process of PN1, PH1 and PK1 specimens shows that this strong interlocking enforces the crack’s development around the strong interfacial bonding area towards the defect region or the weak interfacial area between the fibers and the matrix. When the interlocking of the crack tip is strong, the crack may produce small branches or even new cracks, which increases the curvature of the crack path, resulting in the modified specimens having a greater residual bearing capacity and fracture energy than specimens before modification. The fiber modification can significantly improve the flexural and cracking resistance of the specimens.

To quantitatively assess the effect of fiber modification on the cracking pattern of cemented soil, Abdul et al. [37] proposed a main crack tortuosity ratio, which was defined as the ratio of the main crack path length over the straight height of the specimens. Table 5 lists the main crack zigzag ratios of cemented soil with fiber reinforcement. It can be observed that the major crack curvature of the modified fibers remains larger than that before modification. The KH550-modified fibers have the greatest major crack curvature, which is increased by 21.27% when compared with that before the modification. It can be seen that the modified fibers extended the cracking path and played a role in preventing fracture.

#### 3.2.4. Crack Propagation Rate

Figure 10 shows the crack propagation rates of cemented soil beams reinforced by the fibers modified with varied methods and varied fiber contents. To describe the relationship between crack opening width and time, the parameters of crack propagation rate, *a* (i.e., slope) and initial crack width, *b* (i.e., intercept) can be obtained by a linear fitting with an equation of *y* = *ax* + *b*. The fitting results are given in Table 6. The results show that the fiber crack propagation rate of modified fibers is lower than that before modification. For example, when the fiber content is 1%, the crack propagation rate of the specimens before modification was 0.00175 mm/s, and the crack propagation rate of the specimens modified by NaOH, HCl and KH550 were 0.00165 mm/s, 0.00171 mm/s and 0.0017 mm/s, respectively, which are 6.06%, 2.33% and 2.94% lower than those before modification. Except for specimen PH0.5, the initial crack widths of the modified-fiber-reinforced specimens were smaller than those before modification. PN1, PH1 and PK1 decreased by 0.09 mm, 0.05 mm and 0.03 mm in crack width, respectively, when compared with PVA1, indicating that the modified fibers could effectively transfer stress through the fiber-matrix interface, and thus delay the strain growth and suppress the crack propagation rate.

### 3.3. Fracture Toughness Prediction Model

Fracture energy and peak strength of the specimens are show in Table 7, and Figure 11 shows the relationship between fracture energy and peak strength of the fiber-reinforced cement specimens, from which it can be seen that the fracture energy of PVA fiber increases with the increase in peak strength, and this trend is consistent with the description of the previous fracture parameters. The fracture toughness prediction model was established by regression analysis, which considered the bending strength of the fiber as the independent variable. Fitting parameters of PVA fibers are show in Table 8, where the regression coefficients *R*^2^ of PVA, AN, AH, and AK were 0.9418, 0.9420, 0.9812, and 0.9970. It can be seen that the fracture energy has a strong linear relationship with the peak strength, so the peak strength of the specimens before and after modification can be predicted by the peak strength. This provides a theoretical reference for the engineering applications of fiber-reinforced cemented soil specimens.

## 4. Mechanistic Study

### 4.1. Fiber Surface Morphology

Figure 12 shows that the surface of untreated PVA fiber is relatively smooth, while the surface of PN and PH fibers are longitudinally damaged, with long dents and exfoliations adhering to the fiber surface. According to the mechanical adhesion theory, the modified rougher surface increases the aspect ratio, and thus improves the mechanical interlocking and provides additional anchorage points between the fibers and the cemented soil. Therefore, the interfacial adhesion between the fibers and the matrix is increased [38,39,40]. In contrast, a hydrophilic coating is formed on the surface of KH550-treated fibers, which fills the space between the original fibers, so that the fiber surface is still smooth.

### 4.2. Pore Structure

In order to better explain the mechanism of bending in the reinforced cemented soil with modified fibers, the fracture surface of the specimens was taken for mercury injection tests. Different fiber modification methods lead to significant differences in the pore structure of the specimens. The alterations in pore structure are the most direct expression that affects the macroscopic behavior of the composite. The coarser pores promote the initiation and propagation of cracks, and thus promote a rapid load drop, causing failure of the specimen. The cumulative pore volume and differential pore size distribution of specimens are shown in Figure 13. Li et al. [41] concluded that the pore size of specimens could be classified into four ranges: microporous (<10 nm), small (10–100 nm), mesopores (100–1000 nm) and large (>1000 nm). It can be observed that the fiber treatment changes the pore proportion in the specimens: the large pores are crushed into small pores, in other words, and the number of small pores increases, while the number of mesopores and large pores decreases. This indicates that the fiber modification refines the pore structure, which leads to a denser microstructure and improves the interfacial bonding effect between the fiber and the matrix. This observation is in good agreement with the bending test results of Wu et al. [42].

## 5. Conclusions

In this paper, the effects of fiber modification methods and fiber content on the flexural behavior of cemented soil were investigated by performing three-point bending tests. The flexural strength and fracture energy were analyzed. The DIC technique was applied to obtain the strain field on the specimens’ surface, and thus the crack propagation process was examined during fracture. Form this work, the following conclusions were made:(1)The three fiber modification methods can improve the flexural strength and fracture energy of the reinforced cemented soil, and the peak strength can be used to accurately predict the fracture energy of the specimens.(2)The fiber surface modification can change the crack morphology of cemented soil from a linear shape to a curved shape. This increases the curvature path of the crack, and gradually change its failure mode from brittleness to ductility.(3)The modified fibers can effectively delay the crack propagation rate and reduce the initial crack width. The SEM and mercury injection test show that the fiber modification increases the roughness of the fiber surface and improves the pore structure of the specimens, resulting in a good interfacial bonding between the modified fibers and the matrix.(4)According to the bending parameters and crack propagation process of fiber-reinforced cemented soil, the bending resistance of NaOH-modified fiber specimens is the best, followed by the modification effect of HCl, and finally by KH550.

## Figures and Tables

**Figure 1 materials-15-04799-f001:**
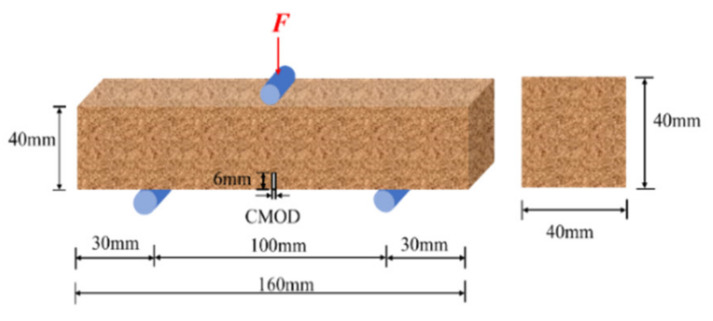
Schematic diagram of size of three-point bending specimen.

**Figure 2 materials-15-04799-f002:**
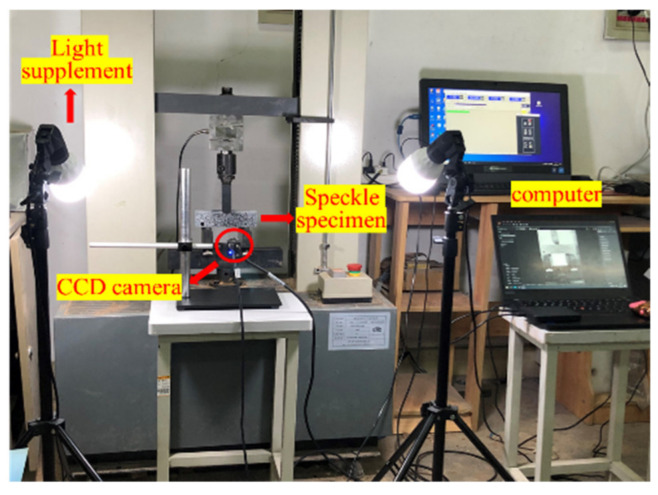
Schematic diagram of DIC test system.

**Figure 3 materials-15-04799-f003:**
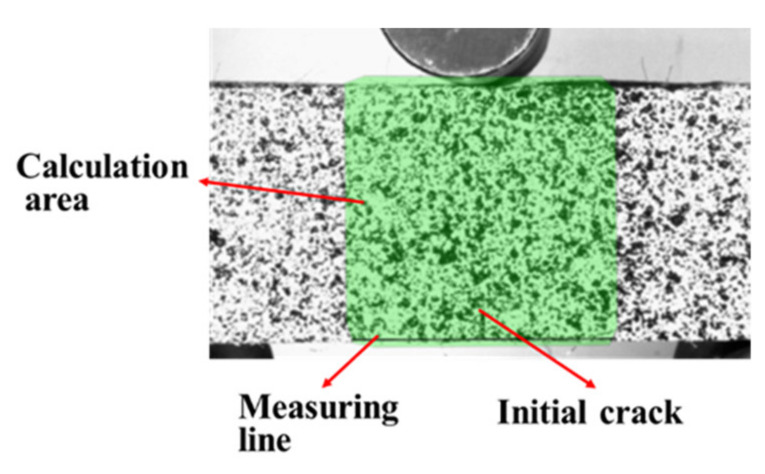
Measurement line of the initial crack area.

**Figure 4 materials-15-04799-f004:**
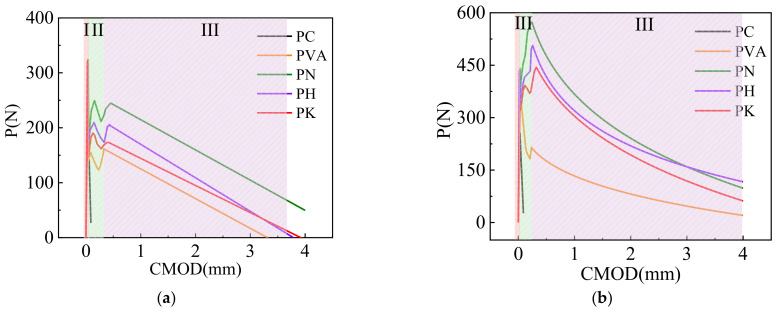
Load-crack mouth opening displacement curves of cemented soil with fiber contents of 0.25% (**a**), 0.5% (**b**) and 1% (**c**) before and after modification.

**Figure 5 materials-15-04799-f005:**
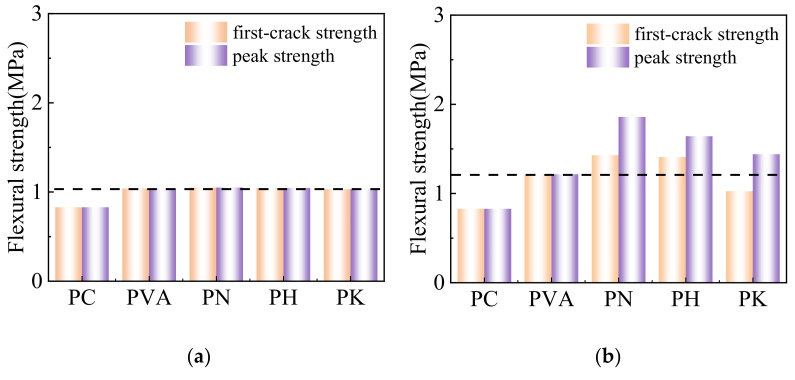
Flexural strength of cemented soil before and after modification with fiber contents of 0.25% (**a**), 0.5% (**b**) and 1% (**c**).

**Figure 6 materials-15-04799-f006:**
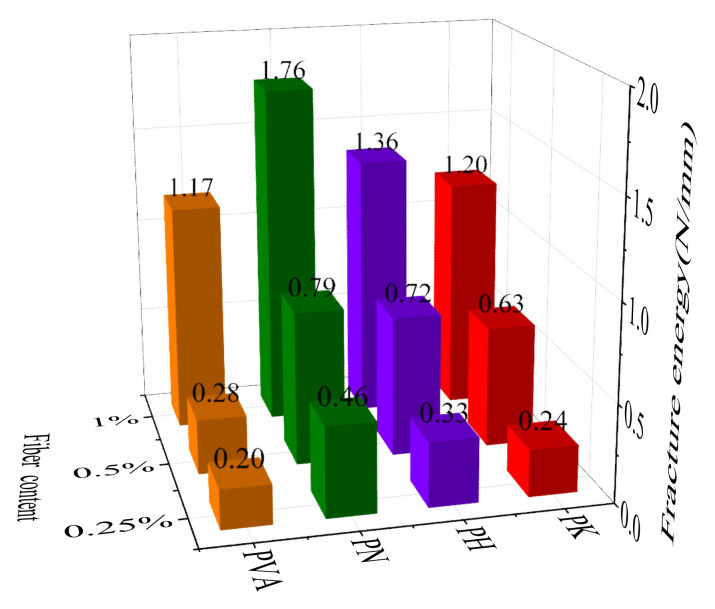
Fracture energy of cemented soil before and after modification.

**Figure 7 materials-15-04799-f007:**
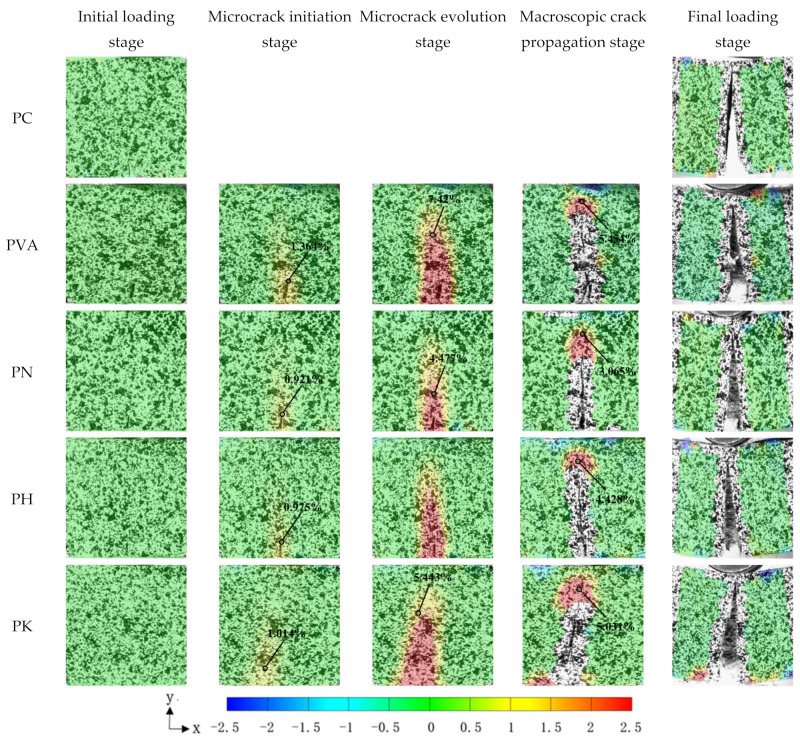
Strain field along X direction of the specimens with 1% fiber content.

**Figure 8 materials-15-04799-f008:**
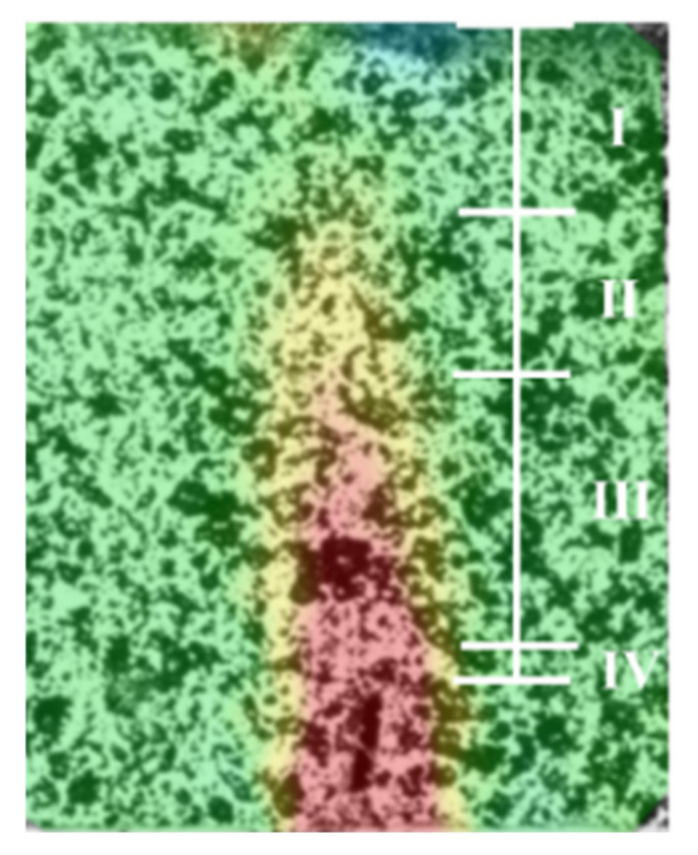
Crack tip region.

**Figure 9 materials-15-04799-f009:**
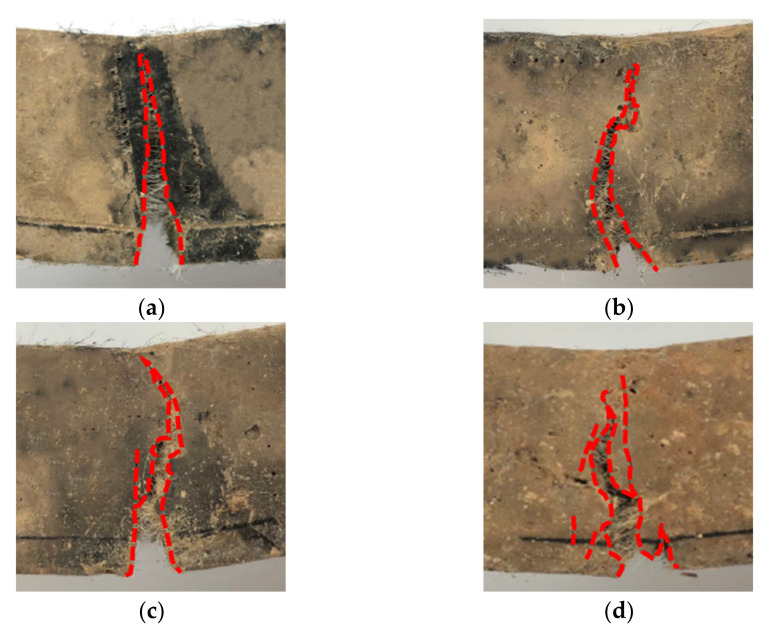
Final crack path of notched specimens with 1% fiber content specimens. (**a**) PVA; (**b**) AN; (**c**) AH; (**d**) AK.

**Figure 10 materials-15-04799-f010:**
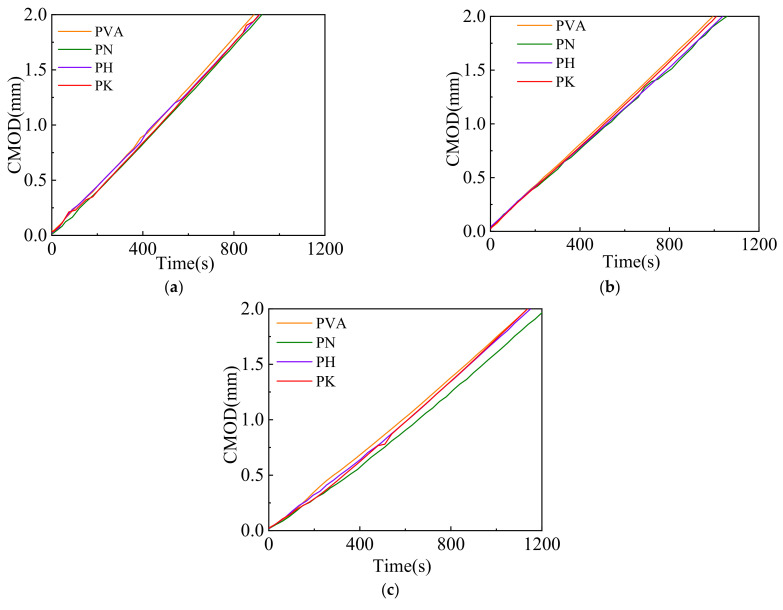
Crack propagation rates of cemented soil before and after modification with fiber content of 0.25% (**a**), 0.5% (**b**) and 1% (**c**).

**Figure 11 materials-15-04799-f011:**
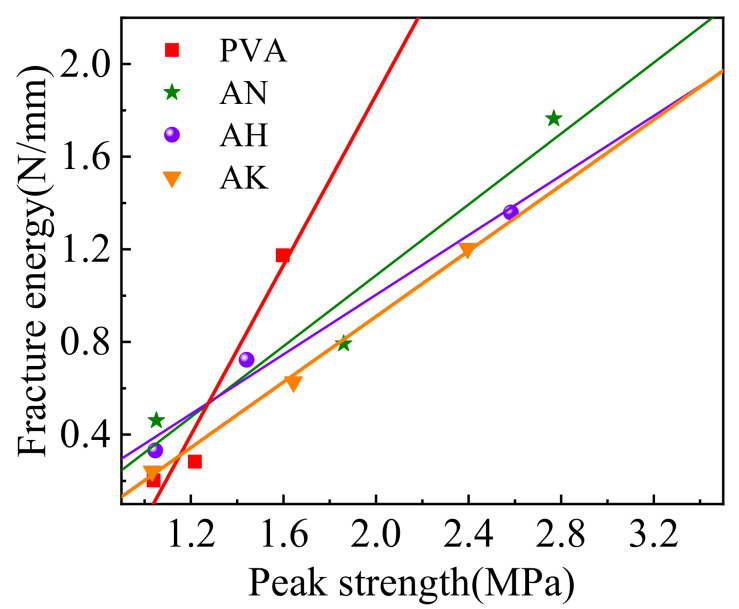
Relationship between fracture energy and peak strength.

**Figure 12 materials-15-04799-f012:**
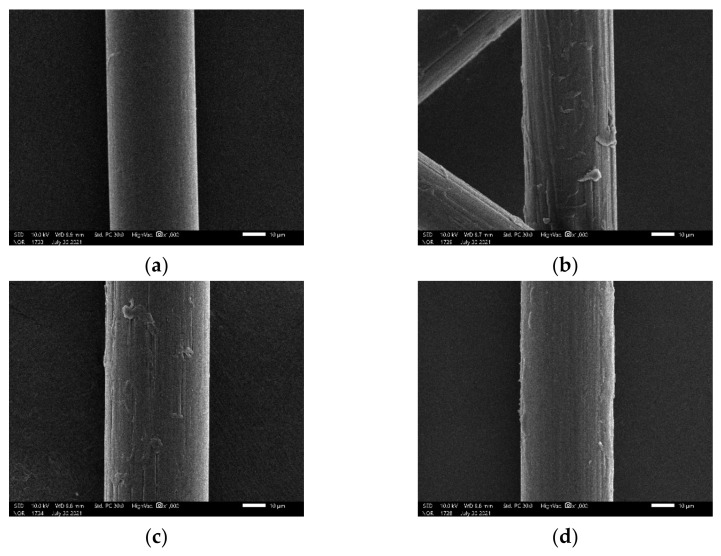
Surface morphology PVA fibers of unmodified (**a**) and NaOH modified (**b**), HCl modified (**c**), KH550 (**d**) modified.

**Figure 13 materials-15-04799-f013:**
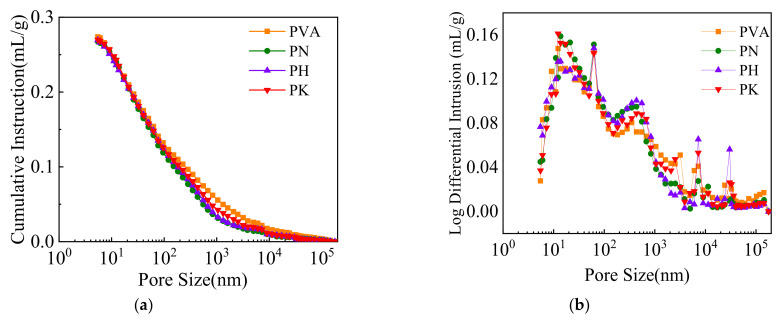
Pore structure of the specimens with 1% fiber content: (**a**) cumulative pore volume, (**b**) pore size distribution.

**Table 1 materials-15-04799-t001:** Soil properties.

Soil Properties	Specific Gravity	Moisture Content (%)	Dry Density (g/cm^3^)	Liquid Limit (%)	Plastic Limit (%)	Plasticity Index
Value	2.7	6.79	1.82	25.25	15.61	9.6

**Table 2 materials-15-04799-t002:** Fiber properties.

Soil Properties	Specific Gravity (g/cm^3^)	Length (mm)	Aspect Ratio (μm)	Tensile Strength (MPa)
Value	1.3	12	40	1600

**Table 3 materials-15-04799-t003:** PVA fiber surface modification.

Surface Modification Method	Modified Concentration	Modification Times
Alkali treatment (NaOH)	5%	6 h
Acid treatment (HCl)	5%	6 h
Silane coupling agent (KH550)	1%	6 h

**Table 4 materials-15-04799-t004:** Mixture proportions.

Specimen Legend	Dry Soil (g)	Cement (%)	The Water Content (%)	Fiber Content (%)
PC	100	15	33	0
PVA0.25	100	15	33	0.25
PN0.25	100	15	33	0.25
PH0.25	100	15	33	0.25
PK0.25	100	15	33	0.25
PVA0.5	100	15	33	0.5
PN 0.5	100	15	33	0.5
PH0.5	100	15	33	0.5
PK0.5	100	15	33	0.5
PVA1	100	15	33	1
PN1	100	15	33	1
PH1	100	15	33	1
PK1	100	15	33	1

Notes: (1) Water content, which is defined as the liquid limit of test soil plus 0.5 water–cement ratio, is 32.75, while a value of 33 is taken 33. (2) Fiber content is the ratio of fiber weight to soil dry weight. (3) PC-cemented soil, PVA-unmodified specimen, PN modified by NaOH, PH modified by HCl, PK modified by KH550.

**Table 5 materials-15-04799-t005:** Main crack tortuosity of specimens with 1% fiber content.

Specimens	Main Crack Length (mm)	Primary Crack Tortuous Rate
PVA	371.515	1.011
PN	396.949	1.198
PH	434.055	1.170
PK	426.733	1.226

**Table 6 materials-15-04799-t006:** Fitting parameters of crack propagation process.

Specimens	0.25% Fiber Content		0.5% Fiber Content		1% Fiber Content	
	a	b	a	b	a	b
PVA	0.00226	0.042	0.00199	0.028	0.00175	0.025
PN	0.00220	0.028	0.00189	0.019	0.00165	0.01
PH	0.00218	0.025	0.00190	0.022	0.00171	0.018
PK	0.00221	0.033	0.00196	0.025	0.00170	0.022

**Table 7 materials-15-04799-t007:** Fracture energy and peak strength of the specimens.

Specimens	Fiber Content 0.25%		Fiber Content 0.5%		Fiber Content 1%	
	Peak Strength	Fracture Energy	Peak Strength	Fracture Energy	Peak Strength	Fracture Energy
PVA	1.038	0.202	1.218	0.283	1.597	1.174
PN	1.051	0.461	1.861	0.793	2.768	1.764
PH	1.046	0.331	1.441	0.723	2.583	1.359
PK	1.032	0.241	1.644	0.627	2.396	1.184

**Table 8 materials-15-04799-t008:** Fitting parameters of PVA fibers.

Specimens	Relationship Formula	*R* ^2^
PVA	*y* = 1.84*x* − 1.80	0.9418
AN	*y* = 0.77*x* − 0.44	0.9420
AH	*y* = 0.64*x* − 0.28	0.9812
AK	*y* = 0.71*x* − 0.51	0.9970

## Data Availability

The data used to support the findings of this study are available from the corresponding author upon request.

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
