# Peer review of "Bending and Crack Evolution Behaviors of Cemented Soil Reinforced with Surface Modified PVA Fiber"

_materials, 2022, doi:10.3390/ma15144799_

Round 1
Reviewer 1 Report
This paper presents an interesting study about the mechanical properties of cemented soil reinforced with surface modified PVA fiber. In general, the bending and crack evolution performance of such a material are explored. The results presented by the Authors are pretty solid, however, the Introduction Section must be improved. Finally, I am summarizing my main observations and recommendations that the Authors must address before proceeding further.
- In the Abstract Section, the following sentence must be rephrased “In this study, how different fibre surface modifications, i.e. alkali treatment, acid treatment and silane coupling agent treatment, as well as different fibre contents, i.e. 0%, 0.25%, 0.5% and 1% affect the bending properties of cemented soils was investigated by conducting the three-point bending tests on notch beams.”. In general, such a sentence is very long, I would recommend to the Authors splitting it in two or three sentences.
- In some parts of the manuscript, it seems that the Authors are using the word “fibre”, I am not quite sure if this is correct. From my point of view, it must be “fiber” instead of “fibre”. Please revise this.
- By the end of the Abstract Section, the Authors declared the SEM acronym. However, the meaning of SEM is not reported previously in the paper. Please report the meaning of SEM in the Abstract Section.
- By the End of the Abstract Section, in one or two lines, please declare the main findings of the paper. This will attract the attention of possible readers.
- In the Introduction Section, first paragraph, last sentence, please change “soil matrixis necessary to be explored[7].” to “soil matrix is necessary to be explored [7].”.
- The Introduction Section must be improved. It seems that only 11 references are discussed in such a section. Please increase the number of references used to develop the state of the art of the problem that will be addressed in the paper. Remember that the Introduction Section must clearly describe what have been done in the past about the topic of the paper and reflects the main shortcomings of other investigations. Please consider this to revise the Introduction Section.
- By the end of the Introduction Section, the Authors must develop a clear discussion of the main objective of the paper and the contribution to the Materials journal. I think that the Authors are trying to do so but the idea still being so disperse. Please improve the explanation of the contribution of this paper to the Materials Journal.
- In Section 2.1, the Authors are describing the materials used in the experiment. Please include a real photograph of the PVA fiber used in the experiment.
- Tables and Figures must be incorporated in the manuscript right after they are mentioned in the text. For example, Figure 1 must be incorporated right after the paragraph where it was reported. This is not followed in the manuscript. Figure 1 is referred in page 2 of the manuscript and is incorporated in the text up to page 4. Please revise this type of problems.
- Are mixture proportions reported in Table 4 recommended in the literature? Or are they only proposed by the Authors? Please justify this.
- Please include a picture where the speckle specimen can be noticed. In Figure 1 it is marked but is very far away to see it properly.
- By the end of Section 2, please include a flowchart describing the process of the experimental part of the paper.
- Please include a list of acronyms.
- In some part of the paper, please discuss some real examples where cemented soil reinforced with surface modified PVA fiber has been implemented as a construction material (e.g., roads, pavements, etc.).
- Based on the results presented in the paper. The following question must be answered. Up to this stage in the state of the art, can cemented soil reinforced with surface modified PVA fiber be used as a reliably material in the construction industry? Please justify your answer in the manuscript.
- In the Conclusions Section of the paper, please include one more conclusion about the main limitations of the study performed in the paper.
- In the References Section, please include DOI to every reported paper.
Author Response
Response to Reviewer 1 Comments
Dear Editor and Reviewers:
Firstly, we would like to thank you for your kind letter and for reviews’ constructive comments concerning our article (Manuscript ID: materials-1632399). These comments are all valuable and helpful for improving our writing. We feel that the comments have greatly benefited the manuscript and are grateful for you time.
We have tried our best to revise the manuscript according to your kind and construction comments and suggestions. Each suggested revision and comment, brought forward by the reviewers was accurately incorporated and considered. We have provided a point-by-point response to the reviewers' comments below in red color. We sincerely hope that this revised manuscript has addressed all your comments and suggestions. The following is the highlighted revisions that we have made:
Point 1: In the Abstract Section, the following sentence must be rephrased “In this study, how different fiber surface modifications, i.e. alkali treatment, acid treatment and silane coupling agent treatment, as well as different fiber contents, i.e. 0%, 0.25%, 0.5% and 1% affect the bending properties of cemented soils was investigated by conducting the three-point bending tests on notch beams.”. In general, such a sentence is very long, I would recommend to the Authors splitting it in two or three sentences.
Response 1:
We appreciate it very much for this good suggestion, “In this study, how different fibre surface modifications, i.e. alkali treatment, acid treatment and silane coupling agent treatment, as well as different fibre contents, i.e. 0%, 0.25%, 0.5% and 1% affect the bending properties of cemented soils was investigated by conducting the three-point bending tests on notch beams.”
In this paper, the effects of fiber surface modification methods and fiber content on the bending properties of cemented soils were investigated by conducting the three-point bending tests on notch beams. Among them, the fiber surface modification methods are alkali treatment, acid treatment, silane coupling agent treatment, fiber content is 0%, 0.25%, 0.5%, 1%.
Point 2: In some parts of the manuscript, it seems that the Authors are using the word “fibre”, I am not quite sure if this is correct. From my point of view, it must be “fiber” instead of “fibre”. Please revise this.
Response 2:
It is really a good idea as Reviewer suggested, we already replace all the words “fibre” in this article with “fibre”.
Point 3: By the end of the Abstract Section, the Authors declared the SEM acronym. However, the meaning of SEM is not reported previously in the paper. Please report the meaning of SEM in the Abstract Section.
Response 3:
Thank you so much for your careful check, the meaning of SEM is Scanning Electron Microscope.
Point 4: By the End of the Abstract Section, in one or two lines, please declare the main findings of the paper. This will attract the attention of possible readers.
Response 4:
The results show that the fiber surface modification will not significantly improve the elastic deformation and initial crack strength of the cemented soil, but can greatly improve the residual strength, ductility and energy absorption capacity of the specimens and gradually change their damage mode from brittle to ductile. In addition, the strain at the crack tip, the crack propagation rate and initial crack width of the modified specimens were lower than those before modification, indicating that the bonding performance of the modified fibers to the matrix was better, which could effectively transfer stress through the fiber-matrix interface, thus retarding the crack propagation and improving the strain resistance and deformation resistance of the specimens.
Point 5: In the Introduction Section, first paragraph, last sentence, please change “soil matrixis necessary to be explored[7].” to “soil matrix is necessary to be explored [7].”.
Response 5:
We already changed “soil matrixis necessary to be explored.” to “soil matrix is necessary to be explored.”.
Point 6: The Introduction Section must be improved. It seems that only 11 references are discussed in such a section. Please increase the number of references used to develop the state of the art of the problem that will be addressed in the paper. Remember that the Introduction Section must clearly describe what have been done in the past about the topic of the paper and reflects the main shortcomings of other investigations. Please consider this to revise the Introduction Section.
Response 5:
The content of the introduction has been supplemented, and the specific content can be found in the text, in which 27 references are discussed.
Related research shows that fiber surface modification can increase the roughness of the fiber surface, improve the fiber surface energy, and achieve surface activation, thus increasing the force between the fiber and the matrix and improving the interfacial bonding properties between the fiber and the matrix. But most of the previous studies focus on the fiber surface modification of reinforced epoxy resin, concrete and other composite materials, and the focus of the research is mainly on the mechanical properties of fiber-modified reinforced cementitious composites, and there are few reports on the crack propagation behavior during fracture and engineering applications, and there are few studies on the application of fiber modification to the improvement of flexural properties of cement retaining walls.
Point 7: By the end of the Introduction Section, the Authors must develop a clear discussion of the main objective of the paper and the contribution to the Materials journal. I think that the Authors are trying to do so but the idea still being so disperse. Please improve the explanation of the contribution of this paper to the Materials Journal.
Response 7:
In this paper, NaOH solution, HCl solution and KH550 solution were used to modify the fiber surface and prepare the fiber-cement composite with good interfacial bonding properties, and the effect of fiber surface modification on the bending properties of cementitious materials and the change law were systematically studied, and the mechanism of fiber surface modification by different solutions was elucidated, which expanded the application of fiber-cementitious materials as stirred piles in foundation pit enclosure, and enriched and developed the interfacial bonding theory of fiber reinforced cementitious materials.
Point 8: In Section 2.1, the Authors are describing the materials used in the experiment. Please include a real photograph of the PVA fiber used in the experiment.
Response 8:
Fig. 1 PVA fiber blowing dispersion
Point 9: Tables and Figures must be incorporated in the manuscript right after they are mentioned in the text. For example, Figure 1 must be incorporated right after the paragraph where it was reported. This is not followed in the manuscript. Figure 1 is referred in page 2 of the manuscript and is incorporated in the text up to page 4. Please revise this type of problems.
Response 9:
Tables and diagrams already included immediately after they are mentioned in the text. Please refer to the text for details.
Point 10: Are mixture proportions reported in Table 4 recommended in the literature? Or are they only proposed by the Authors? Please justify this.
Response 10:
The study of Liu et al. shows that sand has high compressive strength when the cement content is 15%. The water content and fiber content were based on Xin et al. references.
- Liu J, An R, Jiang Z, et al. Effects of w/b ratio, fly ash, limestone calcined clay, seawater and sea-sand on workability, mechanical properties, drying shrinkage behavior and micro-structural characteristics of concrete[J]. Construction and Building Materials, 2022,321:126333.
- Xin Yao; Kaixiang Liu; Geng Huang; Mingming Wang; Xiaoqiang Dong. Mechanical properties and durability of deep soil–cement column reinforced by Jute and PVA fiber. J Mater Civil Eng, 2021, 3(4): 4333-4338.
Point 11: I Please include a picture where the speckle specimen can be noticed. In Figure 1 it is marked but is very far away to see it properly.
Response 11:
Fig. 2 Speckle image of specimen surface
Point 12: By the end of Section 2, please include a flowchart describing the process of the experimental part of the paper.
Response 12:
Fig. 3 Technical roadmap
Point 13: Please include a list of acronyms.
Response 13:
|
acronyms |
the full name |
|
SEM |
scanning electron microscope |
|
DIC |
digital image correlation |
|
CCD |
charge-coupled device |
|
LED |
light emitting diode |
|
CMOD |
crack mouth opening displacement |
Point 14: In some part of the paper, please discuss some real examples where cemented soil reinforced with surface modified PVA fiber has been implemented as a construction material (e.g., roads, pavements, etc.).
Response 14:
At present, most of the fiber surface modification technology is applied to strengthen epoxy resin, concrete and other composite materials, and the focus of research is mainly on the mechanical properties of fiber-modified reinforced cementitious composites, the PVA fiber surface modification applied to engineering examples of research has rarely been reported.
Point 15: Based on the results presented in the paper. The following question must be answered. Up to this stage in the state of the art, can cemented soil reinforced with surface modified PVA fiber be used as a reliably material in the construction industry? Please justify your answer in the manuscript.
Response 15:
Cemented soil reinforced with surface modified PVA fiber can be used as a reliably material in the construction industry.
(1) Fibers are widely used in critical parts of structural engineering for earthquake resistance (beams, columns, walls, etc.), bridge decks and pavement (road and airport runways) surfaces, mining and tunneling projects, concrete repair and reinforcement, and various concrete materials, and all have achieved some success.
(2) Fiber surface modification technology can improve the surface roughness and specific surface area of fibers, increase the mechanical force between fibers and matrix, and at the same time, increase the active elements and functional groups on the fiber surface, regulate and optimize the interfacial bonding properties of fibers and matrix, thus improving the mechanical properties of composite materials.
(3) The fiber surface modification method selected in this paper has the advantages of low price, simple operation and no mechanical equipment, so it can be widely used in actual construction.
Point 16: In the Conclusions Section of the paper, please include one more conclusion about the main limitations of the study performed in the paper.
Response 16:
Fiber reinforcement technology is not applicable in all situations, and this document describes some of its limitations. Because of the hygroscopic nature of fibers, it is difficult to apply them under conditions of low moisture content. In addition, The fibers tend to agglomerate during construction, so the construction sequence and process may need to be adjusted and optimized to ensure uniform and dispersed incorporation of the fibers into the cemented soil.
Point 17: In the References Section, please include DOI to every reported paper.
Response 17:
Thank you for your suggestions. I checked the format of the references of this journal on the official website again and found that the references of this journal already included basic information, so the DOI of the paper are not necessary add after each reference.
We sincerely hope that this revised manuscript has addressed all your comments and suggestions. We appreciated for reviewers’ warm work earnestly and hope that the correction will meet with approval.
Once again, thank you very much for your positive and constructive comments and suggestions on our manuscript. If there are any problems, please do not hesitate to contact us.
Best regards,
Lisheng Liang

Reviewer 2 Report
The manuscript entitled: ‘Bending and Crack Evolution Behaviours of Cemented Soil Reinforced with Surface Modified PVA Fiber’ is in line with the Materials journal. It is based on original research. The article is composed properly. The presented research is well designed and analyzed, but requires additional discussion. Moreover, the article requires intensive editing work, including:
- Over the title: follow the journal template – add the type of article.
- Affiliations: corresponding author is required.
- Abstract: add some measurable results.
- All text: requires editing and English correction.
- Introduction: please stress the novelty aspects in presented research.
- Chapter 2.4: please add manufacturer for the UTS.
- Figure 6: Could you change the way of presentation, more vertical chart?
- Discussion: please add discussion and comparison of obtained results with the state-f-the -art, especially literature (IMPORTANT!)
- COI, author contribution and other: please follow the journal template – add lacking information.
- References: please follow the journal template – style editing is needed.
Author Response
Response to Reviewer 2 Comments
Dear Editor and Reviewers:
Firstly, we would like to thank you for your kind letter and for reviews’ constructive comments concerning our article (Manuscript ID: materials-1632399). These comments are all valuable and helpful for improving our writing. We feel that the comments have greatly benefited the manuscript and are grateful for you time.
We have tried our best to revise the manuscript according to your kind and construction comments and suggestions. Each suggested revision and comment, brought forward by the reviewers was accurately incorporated and considered. We have provided a point-by-point response to the reviewers' comments below in red color. We sincerely hope that this revised manuscript has addressed all your comments and suggestions. The following is the highlighted revisions that we have made:
Point 1: Over the title: follow the journal template – add the type of article.
Response 1:
The type of this article is research, and the logo above the title is added by the editor after acceptance.
Point 2: Affiliations: corresponding author is required.
Response 2:
The corresponding author and the first author of this paper are Lisheng Liang, and the corresponding author has been highlighted.
Point 3: Abstract: add some measurable results.
Response 3:
The peak flexural strengths of AN1, AH1 and AK1 specimens were increased by 73.3%, 61.7% and 49.9%, respectively, compared to PVA1. In addition, the fracture energy of specimens AN1, AH1 and AK1 were increased by 180.4%, 121.5% and 155.4%, respectively, compared to PVA1, which showed that the bending properties of NaOH-modified PVA fiber specimens were better.
Point 4: All text: requires editing and English correction.
Response 4:
Thank you for your very useful suggestions. The full text has been polished. Please see the text for detailed modifications.
Point 5: Introduction: please stress the novelty aspects in presented research.
Response 5:
In this paper, we investigate the effects of fiber modification methods and the mixing effect of fiber admixture on the flexural behavior of cement by three-point bending tests of prefabricated incised beams, and the main innovations are as follows.
(1) Application innovation: Most of the current research focuses on the mechanical properties of composite materials such as fiber surface modification enhanced epoxy resin, mortar and concrete, while the research on fiber surface treatment to improve the flexural properties of hydric soil is almost in a blank state. And the cement soil retaining wall is a typical form of foundation support in soft soil areas, when it is subjected to horizontal load will generate bending moment, leading to failure in tension (due to brittleness), so the results of this paper can provide a reference for the reasonable and efficient use of PVA fibers in the field of geotechnical engineering.
(2) Innovation in research direction: the bending behavior of the cement soil specimen is closely related to the crack expansion process (especially in the post-cracking stage), so it is imperative to study the crack expansion process and the evolution of crack tip strain in the specimen. However, previous studies on soil bending performance mainly focus on the quantitative analysis of bending parameters (bending strength and toughness), but the displacement and strain fields on the specimen surface during loading and the dynamic evolution of cracks are rarely involved, so this paper helps researchers to fully understand the bending behavior of cement soil specimens.
(3) Innovative research method: This paper obtains the crack opening width during the loading process of the specimen by analyzing the images obtained by digital image correlation (DIC) technique, while previous studies mostly use electronic extensometer to measure the crack opening width of notched beams, compared with the simple operation and high accuracy of DIC technique.
Point 6: Chapter 2.4: please add manufacturer for the UTS.
Response 6:
The manufacturer of UTS is Shanghai Yihuan Instrument Technology Co.
Point 7: Figure 6: Could you change the way of presentation, more vertical chart?
Response 7:
Change Figure 6 to the following form:
Fig. 6 Fracture energy of cemented soils before and after modification
Point 8: Discussion: please add discussion and comparison of obtained results with the state-f-the -art, especially literature (IMPORTANT!)
Response 8:
The results:
1) Peak flexural strength: The peak flexural strengths of specimens AN1, AH1 and AK1 increased by 73.3%, 61.7% and 49.9%, respectively, compared to PVA1.
2) Residual flexural strength: The residual flexural strength fR1 of specimens AN1, AH1 and AK1 increased by 98.4%, 81.5% and 71.4%, respectively, compared with that before modification. fR2, fR3 and fR4 showed the best residual flexural strength after NaOH modification, which increased by 59.6%, 47.4% and 11.4%, respectively, compared with that before modification.
(3) Fracture energy: the fracture energy of specimens AN1, AH1 and AK1 increased by 180.4%, 121.5% and 155.4%, respectively, compared with that before modification.
Yan et al [1] showed that alkali treatment increased the compressive stress (+7.1%) and strain (+70%) and flexural strength (+21.4%) of the composites.Du et al [2] found that treatment with 2% silane coupling agent KH550 increased the flexural toughness and energy absorption capacity of ultra-high performance fiber reinforced concrete by 47.6% and 78.0%, respectively.
Compared with the experimental data from the literature on the above parameters, the chemical modification method employed in this paper was found to be an efficient solution to improve the interfacial bonding between the fibers and the cementitious composite and to further improve the flexural properties of the composite.
The state-f-the -art:From the above literature, it can be seen that plasma has good effect on fiber surface modification, and its effect is related to factors such as discharge plasma strength, uniformity and treatment time, but a high vacuum environment is required in the actual operation and the uniformity of plasma should be ensured, moreover, the instrumentation of plasma is costly and the operation is complicated, so it is difficult to apply plasma treatment technology in practice. Nanomaterials can only be fully developed if they are uniformly dispersed in the polymer matrix, but because nanoparticles have high surface energy and tend to agglomerate, they can have a detrimental effect on the mechanical properties of the composites, in addition, the cost of nanomaterials is relatively high. In summary, compared with physical modification and combined modification, chemical modification can be widely used in actual construction because of the advantages of low price, simple operation, and no mechanical equipment, so the use of chemical treatment to optimize and regulate the interfacial properties is a more suitable method.
- YAN L, CHOUW N, HUANG L, et al. Effect of alkali treatment on microstructure and mechanical properties of coir fibres, coir fibre reinforced-polymer composites and reinforced-cementitious composites[J]. Construction and Building Materials, 2016,112: 168-182.
- DU S, ZHOU Y, SUN H, et al. The effect of silane surface treatment on the mechanical properties of UHPFRC[J]. Construction and Building Materials, 2021,304: 124580.
Point 9: COI, author contribution and other: please follow the journal template – add lacking information.
Response 9:
Author Contributions: Lisheng Liang: Data Curation, Writing Original Draft, Review; Yaxing Xu: Review, and; Shunlei Hu: Review. All authors have read and agreed to the published version of the manuscript.
Funding: This work was funded by National Natural Science Foundation of China (Grant No. 51978438).
Institutional Review Board Statement:
Informed Consent Statement: Informed consent was obtained from all subjects involved in the study.
Data Availability Statement: The data used to support the findings of this study are available from the corresponding author upon request.
Point 10: References: please follow the journal template – style editing is needed.
Response 10:
The references have been edited according to the journal template, please refer to the text for details.
We sincerely hope that this revised manuscript has addressed all your comments and suggestions. We appreciated for reviewers’ warm work earnestly and hope that the correction will meet with approval.
Once again, thank you very much for your positive and constructive comments and suggestions on our manuscript. If there are any problems, please do not hesitate to contact us.
Best regards,
Lisheng Liang

Reviewer 3 Report
Comments
This paper investigate the Behaviours of Cemented Soil Reinforced with Surface Modified PVA Fiber. The outcome of the paper is interesting however, there are several aspects that need to be improved. The reviewer can only recommend for publication if the author satisfactorily address the following major comments in the revised version.
- It is a bit surprising that all the values (except PC) in Fig. 5a are same. Any explanation for this observation?
- The research gap from the literature review should be clearly presented.
- The research questions and justification of selecting variables should be highlighted.
- Which test standards was considered in this study? How many replicate samples were tested in each category?
- The failure mechanism of the specimen should be discussed more clearly.
- The novelty of the study should be highlighted more clearly at the end of introduction section. How this study is different from the published study in literature?
- How the outcome of this study will benefit researchers and end users? This need to be highlighted in introduction or end of conclusion.
- The application of fibers is interesting but not novel. Therefore, the recent application in this area should be discussed in introduction section to improve the background study. Recently, waste fibres applied in polymer concrete [Ref: Investigation on the physical, mechanical and microstructural properties of epoxy polymer matrix with crumb rubber and short fibres for composite railway sleepers] and cement concrete [Ref: Effect of short fibres in the mechanical properties of geopolymer mortar containing oil-Contaminated sand]. Suggest to include them in introduction section with proper citations to improve the background study.
I would be happy to see the revised version to understand how these comments are being addressed.
Author Response
Response to Reviewer 3 Comments
Dear Editor and Reviewers:
Firstly, we would like to thank you for your kind letter and for reviews’ constructive comments concerning our article (Manuscript ID: materials-1632399). These comments are all valuable and helpful for improving our writing. We feel that the comments have greatly benefited the manuscript and are grateful for you time.
We have tried our best to revise the manuscript according to your kind and construction comments and suggestions. Each suggested revision and comment, brought forward by the reviewers was accurately incorporated and considered. We have provided a point-by-point response to the reviewers' comments below in red color. We sincerely hope that this revised manuscript has addressed all your comments and suggestions. The following is the highlighted revisions that we have made:
Point 1: It is a bit surprising that all the values (except PC) in Fig. 5a are same. Any explanation for this observation?
Response 1:
When the doping amount is 0.25%, the peak flexural strength of PN, PH and PK specimens are 1.051 MPa, 1.046 MPa and 1.033 MPa, so it can be seen that the modified PVA fiber almost does not affect the flexural strength of the cement soil, and the initial cracking strength is equal to the peak strength, the reason is mainly due to the small amount of fiber doping, the distance between the fibers is larger, so the bridging fiber effect is not sufficiently developed to affect the cracking of the matrix, and similar phenomena were observed in studies of concrete or cementitious composites with less fiber content[1].
- Li J, Niu J, Wan C, et al. Comparison of flexural property between high performance polypropylene fiber reinforced lightweight aggregate concrete and steel fiber reinforced lightweight aggregate concrete[J]. Construction and Building Materials, 2017,157:729-736.
Point 2: The research gap from the literature review should be clearly presented.
Response 2:
Most of the previous studies have focused on fiber surface modification reinforced epoxy resin, concrete and other composites, and the focus of the research is mainly on the mechanical properties of fiber-modified reinforced cementitious composites, but little research has been reported on the crack expansion behavior during fracture and engineering applications, and there are few studies on the application of fiber modification to improve the flexural properties of cementitious retaining walls. Therefore, in this paper, the effects of different fiber surface modification methods (alkali treatment, acid treatment, silane coupling agent treatment) and fiber doping amounts (0%, 0.25%, 0.5%, and 1%) on the flexural behavior and strain field properties during fracture of the cement soil were investigated.
Therefore, a total of 25 sets of specimens with different mix ratios were prepared in this paper, and parameters such as load-crack opening width curve, flexural strength and fracture energy of the specimens were analyzed by three-point bending tests of precast notched beams. In addition, digital image correlation (DIC) technique was used to reproduce the crack expansion process of fiber-reinforced cement soil, and a fracture energy prediction model was established by regression analysis, thus providing a theoretical reference for the engineering application of fiber-reinforced cement soil.
Point 3: The research questions and justification of selecting variables should be highlighted.
Response 3:
Lopattananon et al [2] studied the effect of different alkali solution concentrations (1%, 3%, 5% and 7%) on the mechanical properties of pineapple leaf fiber reinforced natural rubber composites and they observed that the tensile properties of 5% NaOH treated specimens were increased by 28%.Mahjoub et al [3] soaked red hemp fibers with 10% and 15% NaOH solutions for 24h and found that The alkali treatment destroys the texture of the fibers and the treated fibers are more twisted, finer and more brittle than the untreated fibers. Therefore the tensile strength of fibers decreases with the increase of NaOH solution concentration and soaking time, so it is not recommended to use high concentration and long time NaOH solution for alkali treatment.Ray et al [4] treated jute fibers in 5% NaOH solution for 0, 2, 4, 6 and 8 h and found that the toughness of fibers increased by 46% after 6 h treatment.
Therefore, the solution of NaOH and HCl was selected as 5% according to the references and the concentration of KH550 solution was set at 1% according to the usage instructions. In addition, the treatment time was 6 h for all three modification methods.
- Lakshmi Narayana V, Bhaskara Rao L. A brief review on the effect of alkali treatment on mechanical properties of various natural fiber reinforced polymer composites[J]. Materials Today: Proceedings, 2021,44:1988-1994.
- Mahjoub R, Yatim J M, Mohd Sam A R, et al. Tensile properties of kenaf fiber due to various conditions of chemical fiber surface modifications[J]. Construction and Building Materials, 2014,55:103-113.
- Ray D, Sarkar B K, Rana A K, et al. Effect of alkali treated jute fibres on composite properties[J]. Bulletin of materials science, 2001,24(2):129-135.
Point 4: Which test standards was considered in this study? How many replicate samples were tested in each category?
Response 4:
In this paper, a three-point bending test was performed according to standard EN14651. During the test, at least three parallel specimens were tested in each group.
Point 5: The failure mechanism of the specimen should be discussed more clearly.
Response 5:
Since friction between the modified fibers and the cement matrix plays a dominant role after debonding, and the surface of the fibers modified by NaOH and HCl has longitudinal indentations, resulting in increased roughness and interfacial interlocking forces on the fiber surface, additional mechanical anchoring effect is provided along the embedded length of the fibers and prevents the relative sliding of fibers in the cement matrix, so the bonding between the fiber surface and the cement matrix is improved. The KH550-modified fibers may improve the interfacial adhesion properties of PVA fibers in the hydromulch matrix due to the reduced wettability of their surfaces, which can produce more hydration products and densify the interfacial transition zone.
In addition, the pore refinement of the fiber-modified specimens was indicated by the mercury-pressure test, and their average pore size was reduced, suggesting that the fiber modification improved the compactness of the fiber-cemented soil matrix, leading to a denser microstructure and better adhesion properties, which effectively inhibited the sprouting and expansion of microcracks, which was in good agreement with the results of the bending performance test.
Point 6: The novelty of the study should be highlighted more clearly at the end of introduction section. How this study is different from the published study in literature?
Response 6:
In this paper, we investigate the effects of fiber modification methods and the mixing effect of fiber admixture on the flexural behavior of cement by three-point bending tests of prefabricated incised beams, and the main innovations are as follows.
(1) Application innovation: Most of the current research focuses on the mechanical properties of composite materials such as fiber surface modification enhanced epoxy resin, mortar and concrete, while the research on fiber surface treatment to improve the flexural properties of hydric soil is almost in a blank state. And the cement soil retaining wall is a typical form of foundation support in soft soil areas, when it is subjected to horizontal load will generate bending moment, leading to failure in tension (due to brittleness), so the results of this paper can provide a reference for the reasonable and efficient use of PVA fibers in the field of geotechnical engineering.
(2) Innovation in research direction: the bending behavior of the cement soil specimen is closely related to the crack expansion process (especially in the post-cracking stage), so it is imperative to study the crack expansion process and the evolution of crack tip strain in the specimen. However, previous studies on soil bending performance mainly focus on the quantitative analysis of bending parameters (bending strength and toughness), but the displacement and strain fields on the specimen surface during loading and the dynamic evolution of cracks are rarely involved, so this paper helps researchers to fully understand the bending behavior of cement soil specimens.
(3) Innovative research method: This paper obtains the crack opening width during the loading process of the specimen by analyzing the images obtained by digital image correlation (DIC) technique, while previous studies mostly use electronic extensometer to measure the crack opening width of notched beams, compared with the simple operation and high accuracy of DIC technique.
Point 7: How the outcome of this study will benefit researchers and end users? This need to be highlighted in introduction or end of conclusion.
Response 7:
In this paper, NaOH solution, HCl solution and KH550 solution are used to modify the fiber surface and prepare the fiber-cement composite with good interfacial bonding properties. The effect of fiber surface modification on the flexural properties and drying shrinkage properties of cementitious materials and the change law are systematically studied, and the mechanism of fiber surface modification by different solutions is elucidated, which expands the application of fiber-cementitious soil as stirred piles in foundation pit enclosure and enriches and develops the interfacial bonding theory of fiber reinforced cementitious materials at the same time.
Point 8: The application of fibers is interesting but not novel. Therefore, the recent application in this area should be discussed in introduction section to improve the background study. Recently, waste fibres applied in polymer concrete [Ref: Investigation on the physical, mechanical and microstructural properties of epoxy polymer matrix with crumb rubber and short fibres for composite railway sleepers] and cement concrete [Ref: Effect of short fibres in the mechanical properties of geopolymer mortar containing oil-Contaminated sand]. Suggest to include them in introduction section with proper citations to improve the background study.
Response 8:
References have been included in the introduction and the introduction has been improved, and its details are given in the text.
We sincerely hope that this revised manuscript has addressed all your comments and suggestions. We appreciated for reviewers’ warm work earnestly and hope that the correction will meet with approval.
Once again, thank you very much for your positive and constructive comments and suggestions on our manuscript. If there are any problems, please do not hesitate to contact us.
Best regards,
Lisheng Liang

Reviewer 4 Report
The paper deals with an interesting topic concerning the surface modification of PVA fibers to reinforce cemented soils. The results could be interesting for application also into concrete.
The paper has a clear structure, but the English language should be improved. Some suggestions are given in the attached file.
Also the impagination, figures and tables, should be improved.
Some questions about the content:
1) You compare the response of the specimens with different fiber surface modification and fibre content in terms of flexural strenght and first-cracking strenght. Please report the formulas you used for notched beams to obtain the strenght from the experimental load P. Moreover, how did you define the load corresponding to the first cracking strength? Is it an objective criteria?
2) The comparisons are also provided in terms of fracture energy. How did you computate the fracture energy? Which formula? Which limit CMOD?
3) In Figure 7 you compare the DIC images for specimens that have the same fiber content but different fibre surface modification. Do the reported strain correspond to the same loading level ?
These aspects should be clarified and properly inserted in the paper for better understanding. Some suggestions are also given in the attached file.

Author Response
Response to Reviewer 4 Comments
Dear Editor and Reviewers:
Firstly, we would like to thank you for your kind letter and for reviews’ constructive comments concerning our article (Manuscript ID: materials-1632399). These comments are all valuable and helpful for improving our writing. We feel that the comments have greatly benefited the manuscript and are grateful for you time.
We have tried our best to revise the manuscript according to your kind and construction comments and suggestions. Each suggested revision and comment, brought forward by the reviewers was accurately incorporated and considered. We have provided a point-by-point response to the reviewers' comments below in red color. We sincerely hope that this revised manuscript has addressed all your comments and suggestions. The following is the highlighted revisions that we have made:
Point 1: You compare the response of the specimens with different fiber surface modification and fibre content in terms of flexural strenght and first-cracking strenght. Please report the formulas you used for notched beams to obtain the strenght from the experimental load P. Moreover, how did you define the load corresponding to the first cracking strength? Is it an objective criteria?
Response 1:
According to the specification EN14651:2005, the following formula is used to calculate the flexural strength of the cement specimen: (1)
Where is the clear span between two supports, taken as 100 mm, is the corresponding load value in the P-CMOD curve, and the first peak load is taken for calculating the initial crack strength, and the maximum load is taken for the peak strength; is the width of the cement specimen, taken as 40 mm; is the height of the cement specimen, taken as 40 mm; and is the width of the prefabricated crack in the cement specimen, taken as 6 mm.
In this paper, the load corresponding to the abrupt change in the displacement field of the specimen surface is defined as the initial cracking load.
It is an objective criteria. According to the DIC principle, when there is a sudden change in the displacement field on the specimen surface, it means that a new crack is sprouting on the specimen surface, so the corresponding load at this time is the initial crack load.
Point 2: The comparisons are also provided in terms of fracture energy. How did you computate the fracture energy? Which formula? Which limit CMOD?
Response 2:
The fracture energy (Gf), called the total fracture energy or specific fracture energy, also known as the resistance to crack expansion, is the energy required to produce a crack per unit of fracture area and can be used to analyze and determine the toughness and resistance to cracking of cement soils, and is also used to characterize the energy dissipation capacity of the material. The fracture energy of the specimen is obtained by performing a three-point bending test on notched beams by specification EN14651 [] thereby, where the fracture energy is calculated by the formula:
(2)
where is the total dissipation work, is the area of the P-CMOD curve,and CMOD is 4mm. is the fracture area, taken as 34 mm × 40 mm.
Point 3: In Figure 7 you compare the DIC images for specimens that have the same fiber content but different fibre surface modification. Do the reported strain correspond to the same loading level ?
Response 3:
The strain in the microcrack initiation stage is the strain at the time of crack initiation on the specimen surface, which is about 90-120 seconds after the start of the test loading.
The strain in the microcrack evolution stage is the strain when the specimen reaches the peak load, which is about 180-240 seconds after the start of the test.
The strain in the macroscopic crack extension stage is the strain at 360 seconds after the test starts loading, and the loading level in this stage is exactly the same.
We sincerely hope that this revised manuscript has addressed all your comments and suggestions. We appreciated for reviewers’ warm work earnestly and hope that the correction will meet with approval.
Once again, thank you very much for your positive and constructive comments and suggestions on our manuscript. If there are any problems, please do not hesitate to contact us.
Best regards,
Lisheng Liang

Round 2
Reviewer 2 Report
The manuscript entitled: ‘Bending and Crack Evolution Behaviours of Cemented Soil Reinforced with Surface Modified PVA Fiber’ had been improved, however it still requires intnsive editing work. Moreover, the discussion is still quite generic.
Author Response
Response to Reviewer 2 Comments
Dear Editor and Reviewers:
Firstly, we would like to thank you for your kind letter and for reviews’ constructive comments concerning our article (Manuscript ID: materials-1632399). These comments are all valuable and helpful for improving our writing. We feel that the comments have greatly benefited the manuscript and are grateful for you time.
We have tried our best to revise the manuscript according to your kind and construction comments and suggestions. Each suggested revision and comment, brought forward by the reviewers was accurately incorporated and considered. We have provided a point-by-point response to the reviewers' comments below in red color. We sincerely hope that this revised manuscript has addressed all your comments and suggestions. The following is the highlighted revisions that we have made:
Point 1: The manuscript entitled: ‘Bending and Crack Evolution Behaviours of Cemented Soil Reinforced with Surface Modified PVA Fiber’ had been improved, however it still requires intnsive editing work. Moreover, the discussion is still quite generic.
Response 1:
Thank you for your very useful suggestions. The full text has been polished. Please see the text for detailed modifications.
We sincerely hope that this revised manuscript has addressed all your comments and suggestions. We appreciated for reviewers’ warm work earnestly and hope that the correction will meet with approval.
Once again, thank you very much for your positive and constructive comments and suggestions on our manuscript. If there are any problems, please do not hesitate to contact us.
Best regards,
Lisheng Liang
Reviewer 3 Report
I have no further comments
Author Response
Dear Editor and Reviewers.
Thank you for your letter and review of our article (manuscript number: Materials-1632399).